# Comprehensive Analysis of Bufadienolide and Protein Profiles of Gland Secretions from Medicinal *Bufo* Species

**DOI:** 10.3390/toxins16030159

**Published:** 2024-03-20

**Authors:** Yunge Fang, Liangmian Chen, Pengfei Wang, Yating Liu, Yuxiu Wang, Zhimin Wang, Yue Ma, Huimin Gao

**Affiliations:** 1National Engineering Laboratory for Quality Control Technology of Chinese Materia Medica, Institute of Chinese Materia Medica, China Academy of Chinese Medical Sciences, Beijing 100700, China; ygyg_fang@163.com (Y.F.); lmchen@icmm.ac.cn (L.C.); fei_wang2024@163.com (P.W.); liuyating963@126.com (Y.L.); xiuxiuxiu8236@163.com (Y.W.); zmwang@icmm.ac.cn (Z.W.); 2School of Chinese Materia Medica, Tianjin University of Traditional Chinese Medicine, Tianjin 301617, China; 3Artemisinin Research Center and Institute of Chinese Materia Medica, China Academy of Chinese Medical Sciences, Beijing 100700, China

**Keywords:** toad venom, *Bufo bufo gargarizans*, *Bufo melanostictus*, *Bufo andrewsi*, *Bufo raddei*, bufadienolides, nano LC-MS/MS

## Abstract

Toad Venom (TV) is the dried product of toxic secretions from *Bufo bufo gargarizans* Cantor (BgC) or *B. melanostictus* Schneider (BmS). Given the increasing medical demand and the severe depletion of wild toads, a number of counterfeit TVs appeared on the market, posing challenges to its quality control. In order to develop an efficient, feasible, and comprehensive approach to evaluate TV quality, a thorough analysis and comparison of chemical compounds among legal species BgC and BmS, as well as the main confusion species *B. andrewsi* Schmidt (BaS) and *B. raddei* Strauch (BrS), were conducted by ultra-performance liquid chromatography–quadrupole time-of-flight mass spectrometry (UPLC-Q-TOF/MS), high performance liquid chromatography (HPLC), sodium dodecylsulfate-polyacrylamide gel electrophoresis (SDS-PAGE), and Nano LC-MS/MS analyses. We identified 126 compounds, including free or conjugated bufadienolides, indole alkaloids and amino acids, among the four *Bufo* species. The content of main bufadienolides, such as gamabufotalin, bufotalin, bufalin, cinobufagin, and resibufogenin, and the total protein contents varied widely among 28 batches of TV due to their origin species. The sum of the five bufadienolides within the BgC, BmS, BaS, and BrS samples were 8.15–15.93%, 2.45–4.14%, 11.15–13.50%, and 13.21–14.68%, respectively. The total protein content of BgC (6.9–24.4%) and BaS (19.1–20.6%) samples were higher than that of BmS (4.8–20.4%) and BrS (10.1–13.7%) samples. Additionally, a total of 1357 proteins were identified. There were differences between the protein compositions among the samples of the four *Bufo* species. The results indicated that BgC TV is of the highest quality; BaS and BrS TV could serve as alternative resources, whereas BmS TV performed poorly overall. This research provides evidence for developing approaches to evaluate TV quality and selecting the proper *Bufo* species as the origin source of TV listed in the Chinese pharmacopoeia.

## 1. Introduction

Toad venom (TV), named Chansu in China, is the dried secretion from the posterior auricular glands or skin glands of *Bufo bufo gargarizans* Cantor (BgC) or *Bufo melanostictus* Schneider (BmS) [1]. As a traditional medicine, TV has been widely used for thousands of years in China, Korea, and Japan in the clinical treatment of heart failure, inflammation, sores, and various cancers [2]. Due to its potent clinical efficacy, in-depth chemical and pharmacological research have been conducted on TV, particularly focusing on two types of micro-molecular components: bufadienolides and indole alkaloids. Over 140 bufadienolides and indole alkaloids have been identified from *Bufo* species, along with sterols and amino acids [2,3]. Additionally, increasing attention has started being paid to the functional proteins and peptides in TV, and a series of active macromolecular components have been identified, including proteins with anti-inflammatory, analgesic, antioxidant, catalytic, and transport activities [4,5,6] as well as polypeptides associated with antitumor and antimicrobial properties [7,8].

In recent years, the increasing demand and environmental damage led to the consequent rise of counterfeit species. Since the appearance of TV does not allow conclusions to be made regarding the species of origin, proper quality control, including the detection of adulteration using the secretion from other *Bufo* species, can only be achieved by chemical analysis [9]. Early investigations were dedicated to the chemical difference of bufadienolides and indole alkaloids in TVs between the legal species, BgC and BmS, using high performance liquid chromatography (HPLC) fingerprinting and liquid chromatography-tandem mass spectrometry (LC-MS/MS) [9,10]. An integrated quality control method, developed by our previous study, utilized characteristic HPLC chromatogram and the quantitative analysis of multi-components by single marker [11,12], which was accepted and listed in the 2020 edition of the Chinese Pharmacopoeia (ChP 2020) [1]. It underscored the batch-to-batch consistency and the convenience of the analytical method, and, to some extent, improved the quality of commercial TVs. However, it has limited performance in identifying adulteration or counterfeiting with other *Bufo* species. Therefore, more attention is needed with regard to the discrepancy of BgC, BmS, and their main counterfeits: besides micro-molecular compounds, the absolute and relative amounts of macromolecular components in TV also need to be considered.

Several reports have shown that proteomic analysis is a helpful approach for the identification of proteins and peptides from the secretion of *Bufo* species [8,13]. A quality control strategy involving two types of components, bufadienolide and protein markers, has been tested and discussed [5,14]. These investigations present the possibility of new approaches to address the challenges of verifying the authenticity of TV and improve the quality control methods. However, so far, few comparative analyses have been published concerning both micromolecules and macromolecules among the TVs from different *Bufo* species.

In this study, 28 batches of TV samples, collected from different *Bufo* species, were comprehensively analyzed, involving legal species (BgC and BmS) and main counterfeits, *B*. *andrewsi* Schmidt (BaS) and *B*. *raddei* Strauch (BrS), from different regions in China. Depending on the characteristics of the chromatograms, totally 126 compounds were identified by ultra-performance liquid chromatography-quadrupole/time-of-flight mass spectrometry (UPLC-Q-TOF/MS), and the differences between *Bufo* species were systematically analyzed. The quantitative analysis of five markers, gamabufotalin (GB), bufotalin (BL), bufalin (BF), cinobufagin (CB), and resibufogenin (RB), was employed to evaluate the TV samples from different *Bufo* species. Additionally, the Bradford method, sodium dodecylsulfate-polyacrylamide gel electrophoresis (SDS-PAGE) and Nano LC-MS/MS were used to identify and analyze the proteins. According to the results, comprehensive evaluations of bufadienolides and proteins in four species were conducted. The representative pictures of four *Bufo* species, their taxonomic relationship as described in *Fauna Sinica* [15], and the workflow of harvesting fresh toad venom and processing TV are shown in Figure 1.

## 2. Results

### 2.1. Qualitative Analysis of Bufadienolides by UPLC-Q-TOF/MS

Twenty-eight batches of TV samples from BgC, BmS, BaS, and BrS were analyzed by UPLC-Q-TOF/MS in positive mode to discover the micro-molecular differences. The base peak ion (BPI) chromatograms of the samples from each species were processed using the fingerprint similarity software (Similarity Evaluation System for Chromatographic Fingerprint of Traditional Chinese Medicine, version 2012.130723). The fitted chromatograms of the four *Bufo* species intuitively visualized the characteristics and the most pronounced difference is shown in Figure 2A. The retention time, accurate molecular mass, molecular formula, and MS/MS fragment ion of compounds was obtained by Masslynx software (version V4.1). Furthermore, the peaks were qualitatively analyzed by comparing with the possible fragmentation pathways reported in the literature [16,17,18] and reference standards. In total, 126 compounds were identified, including 8 amino acids, 6 indole alkaloids, 103 bufadienolides (46 free and 57 conjugated bufadienolides), and 9 unknown components (Table 1). The typical structures and substitutions of micromolecules in TV are shown in Appendix A.

Arginine and histidine were the main amino acids forming diacid-conjugated esters. The diacid-conjugated arginine ester mainly contained C4–C11 diacids and was characterized by the ions [C_8_O_4_N_4_H_12_(CH_2_)_n_]^+^, such as succinyl (*n* = 2, 275 Da), glutaryl (*n* = 3, 289 Da), adipyl (*n* = 4, 303 Da), pimeloyl (*n* = 5, 317 Da), suberoyl (*n* = 6, 331 Da), azelayl (*n* = 7, 345 Da) and sebacyl (*n* = 8, 359 Da). The fragment ions at *m*/*z* 175 indicated the loss of diacid and the retention of arginine fragments according to previous reports [18,19]. And, the ions at *m*/*z* 156 and 170 were identified as the L-histidines and L-1-methylhistidines cleaved from the diacid-conjugated histidine esters [3], respectively.

The alkaloids were characterized by the presence of a central bicylic indole ring, typically with alkyl or alkylamine side chains at the position C-3 of the five-membered ring and a hydroxyl group or sulfate group at C-5 of the benzene ring [19]. All alkaloids produced the same fragment ion at *m*/*z* 160 through the partial fragmentation of the alkylamine side chain (NH_3_, NH_2_CH_3_, NH(CH_3_)_2_, and N(CH_3_)_3_) leaving a propene group. Among the identified alkaloids, serotonin was relatively high in content.

Bufadienolides are a type of steroids with a α-pyrone ring at the 17*β*-position, which consists of free and conjugated bufadienolides. The conjugated bufadienolides were free bufadienolides conjugated with suberoyl arginine esters, dicarboxylic acid hemiesters, or 3-sulfates at the C-3 position [2]. According to the substituent group, bufadienolides can be classified into two types: 14-hydroxyl and 14,15-epoxy substitution. A series of fragmentation ions, such as [M+H]^+^, [2M+H]^+^, [M+H-SO_3_]^+^, [M+H-nH_2_O]^+^, [M+H-nH_2_O-CO]^+^, [M+H-nH_2_O-CH_2_O]^+^, [M+H-CH_2_CO]^+^, [M+H-HOAC]^+^, and [M+H-nH_2_O-C_5_H_4_O_2_]^+^ were clearly observed.

### 2.2. The Comparison between TV Components from Different Species

The response of each identified constituent was extracted from the BPI chromatogram, and the relative abundance of each compound was shown in Table 1. A direct comparison of the MS spectra and the relative abundance of each compound among the four *Bufo* species demonstrated a pronounced difference. The major constituents found in the BgC and BaS samples were generally consistent, including suberoyl arginine (**13**), GB (**25**), arenobufagin (**40**), hellebrigenin (**44**), telocinobufagin (**74**), BL (**79**), cinobufotalin (**91**), BF (**110**), cinobufagin 3-O-suberoyl arginine ester (**122**), CB (**125**), and RB (**126**). The main constituents in BrS samples were suberoyl arginine (**13**), GB (**25**), telocinobufagin (**74**), BL (**79**), marinobufagin (**92**), BF (**110**), and RB (**126**). The BmS samples included suberoyl-L-histidine (**10**), suberoyl-L-1-methylhistidine (**11**), 19-hydroxybufalin (**47**), BL (**79**), marinobufagin (**92**), BF (**110**), RB (**126**), and unknown compounds (**12**, **67**, **95**, **114** and **121**) as the main constituents. For BmS samples, CB (**125**) was detected by MS but mostly undetected by HPLC due to its low abundance. 19-hydroxybufalin was identified as the characteristic component of BmS samples. Compared with three other species (BgC, BaS, and BrS), the conjugated bufadienolides were absolutely dominant in terms of abundance. It was worth noting that high-abundance peaks, such as those of the compounds **67**, **95**, **114,** and **121,** could not be identified structurally based on the limited MS fragmentation. However, the presence of their corresponding fragment ions at *m*/*z* 245 (compound **12**) indicated that these components were derivatives of compound **12**. The isolation and structural elucidation of these components should be conducted in the future.

The mass spectrometry information from 28 samples were imported into the Progenesis QI 2.3 software, and principal component analysis (PCA) was performed to further detect the intrinsic clustering among *Bufo* species samples. The scores and loadings of multivariate analysis were obtained. As shown in Figure 2B, the scores plot of PCA apparently separated all samples into three groups based on their respective origins. In detail, samples from BgC and BaS tended to be classified into one group, while samples from BrS and BmS tended to be classified into the other two separate groups. The results revealed that, considering the fact that the BgC was traditionally recognized as the best source, the micromolecular chemical makeup of BaS was closest to BgC, which was also confirmed by their taxonomic relationships [15]. Nevertheless, the samples from the two legal sources, BgC and BmS, were completely separated. This implies that there are obvious differences between the chemical makeups of these two species, making it difficult to ensure effectiveness and quality.

### 2.3. Quantitative Analysis of Five Marker Bufadienolides

Based on the main common components identified in the qualitative analysis across four species, five bufadienolide markers, GB and BL, along with BF, CB, and RB, were selected as the quantitative markers using a previously developed analytical method [12]. The contents of these five bufadienolides in 28 batches of TV samples are shown in Table 2 and Figure 2C. Representative HPLC profiles are shown in Figure 2D. Among the different samples analyzed, BgC samples exhibited the highest amount of the five bufadienolides in the range of 8.15–15.93%, while BmS samples had the lowest content ranging from 2.45 to 4.14%. The content of the five bufadienolides in BaS samples ranged from 11.15% to 13.50%, and BrS samples showed variations between 13.21% and 14.68%. Notably, the BrS and BaS samples generally displayed higher total amounts of the five bufadienolides compared to other samples examined here. The contents of the five compounds in BaS samples were similar to those in BgC samples. CB was found to be the predominant compound with the highest content both in BgC and BaS samples, while RB and GB served as the primary markers of BrS samples instead. In contrast, RB and BL were identified as the major markers in BmS samples. Considering the ChP 2020 requirement stipulating that the sum of BF, CB, and RB should not be less than 7.0% [1], none of our tested BmS samples met this standard. However, eight batches of BgC samples and all BrS and BaS samples successfully fulfilled this standard.

### 2.4. The Comparison of Bufadienolides from Different Regions in the Same Species

The quality of TV varied among the different species, and the producing areas have a weak impact on quality. Traditionally, Jiangsu and Shandong were regarded as the high-quality producing areas of TV. In this research, a total of six samples collected from Jiangsu and five samples from Shandong were all identified as BgC. The content of five bufadienolides in samples from these two areas was generally higher than that in the samples from other areas. The sample with the highest total content of five bufadienolides was from Jiangsu at 15.93%, while that with the lowest was from Henan at 8.15%. The content of bufadienolides in BmS samples was generally low, but the samples from Fujian showed relatively good quality in the three producing areas of BmS samples. BrS samples from Jilin had a higher bufadienolide content than that from Heilongjiang. Both batches of BaS samples from Sichuan contained higher levels of bufadienolide content. These results indicated that species are the primary contributing factor to TV quality, while the source areas have only a weak influence.

### 2.5. Total Protein Content Determination

The protein component, which is another typical active substance of TV, has been receiving growing attention. Most investigations have focused on the qualitative identification of proteins, and few publications displayed the total protein content of TVs [4,6]. Therefore, the total protein content of 28 TV samples from the four *Bufo* samples was determined by the Bradford method, and the results are shown in Figure 3A and Table 2. The total protein contents of the BgC and BaS samples were higher than those of the BmS and BrS samples. Additionally, the total protein content of the TV samples exhibited significant fluctuation. Among the BgC species, the Gansu sample had the highest total protein content, while the Henan specimens had the lowest (24.4% and 6.9%, respectively). The BmS sample from Fujian had the lowest total protein content at 4.8%.

### 2.6. SDS-PAGE Analysis

Combining the results of the qualitative and quantitative analysis of bufadienolides, alkaloids, and total protein content, the samples from Jiangsu, Guangxi, Sichuan, and Jilin were selected as the representative samples of the four *Bufo* species for SDS-PAGE analysis. The results indicated a common prominent band at 40–55 kDa in the BaS, BrS, and BgC samples, while this band was not observed for BmS samples. A unique band of BmS TV was found outside the range of 180 kDa, which was the key difference from other species (Figure 3B). Meanwhile, little variation was observed in the main bands of BgC and BaS TV, which exhibited common weak bands at 25 kDa and 10–15 kDa along with a relatively intense band at 100 kDa. The BaS TV showed a unique weak band at 130 kDa. In addition to the key band at 40–55 kDa, BrS TV displayed two less intense unique bands at 10 kDa and 25 kDa. Generally, the four *Bufo* species were successfully distinguished by SDS-PAGE analysis in our research. The similarity of protein bands was consistent with the taxonomic relationships between the species [15].

### 2.7. Protein Identification and Bioinformatic Analysis

Proteomic analysis was used to investigate the protein composition of TVs from different species. Based on Nano LC-MS/MS and bioinformatic analysis, a total of 1357 proteins were identified in the representative samples. Among these, 1039 and 917 proteins were identified in BgC-F2 and BgC-1, respectively; 1037 and 1062 proteins were identified in BaS-F1 and BaS-F2, respectively; 1016 proteins were identified in BmS-F1; 777, 746, and 914 were identified in BrS-1, BrS-2, and BrS-3, respectively. These samples contained a total of 527 common proteins among the four species. The molecular weight of all identified proteins ranged from 6.6 to 660.2 kDa, where proteins were relatively abundant within the range of 16–55 kDa. The isoelectric points of all identified proteins ranged from 3.81 to 11.65, focusing mainly on the range of 5.0–10.0. The distribution of proteins based on molecular weights and isoelectric points indicated a similar tendency for different *Bufo* species (Appendix A).

The top 10 proteins in terms of abundance are shown in Appendix A for different *Bufo* samples, and the results indicated that there was variation in the protein compositions among species, with most of them exhibiting binding activity. For example, the src substrate cortactin in BmS and alpha-actinin-1 in BaS were actin filament binding proteins; phosphatidylinositol 5-phosphate 4-kinase type-2 gamma, T-complex protein 1 subunit epsilon in BgC, and ATP-binding cassette sub-family F member 1 in BmS were ATP binding proteins; arylsulfatase D and transcobalamin-2 in BmS, COP9 signalosome complex subunit 3 in BaS, and serotransferrin-A in BrS were metal ion binding proteins. The other top 10 proteins of TV were identical protein binding proteins, calcium ion binding proteins, and heme binding proteins.

The gene ontology (GO) enrichment analysis on all TV proteins was performed on all proteins to investigate the greatest enrichment categories according to the cellular component (CC), molecular function (MF), and biological process (BP). CC categories were mainly enriched in the cytoplasm, cytosol, and proteasome complex. The categories of membrane and organelle were also observed as mentioned in previous studies [5]. In MF, the structural constituent of ribosome, the translation initiation factor activity, and GTP binding were prominent. Catalytic activity and transporter activity were also enriched, which was consistent with previous reports [5]. BP was mainly involved in the translation, protein folding, and multiple energy metabolism process such as glycolytic process and carbohydrate metabolic process. Additionally, the isoprenoid biosynthetic process was observed to participate in the biosynthesis of secondary metabolites. Some BPs that have been reported showed moderate enrichment in this study, including the developmental process, localization, biological regulation, and response to stimulus [4,7]. The top 10 terms of GO enrichment analysis are shown in Figure 3C.

The Kyoto Encyclopedia of Genes and Genomes (KEGG) pathway of proteins identified from four species was enriched, and the most significant pathway was the biosynthesis of antibiotics, which may be related to the defense systems of toads [20]. Other top pathways included the biosynthesis of amino acids, citrate cycle, amino/nucleotide sugar metabolism, fatty acid metabolism, ribosome, proteasome, glycolysis/gluconeogenesis, peroxisome, endocytosis, and protein processing in the endoplasmic reticulum. Some of these pathways were reported by Yang [5]. The top 20 highest significant pathways are shown in Figure 3D.

Furthermore, the identified proteins with potent activities that were not listed in the top 10 were focused on and specifically evaluated in this article. In all *Bufo* species, golgi-associated plant pathogenesis-related protein 1 with antimicrobial activity was observed. Only BgC and BrS samples were found to contain probable E3 ubiquitin-protein ligase makorin-2 with antitumor activity. It is worth noting that certain homologous proteins were identified as distinct entities due to the utilization of different databases [6,21]. For instance, we identified serpin B5 and serpin B8, while serpin B6 was reported previously [5]. Similarly, our analyses revealed glutathione S-transferase Mu 1, glutathione S-transferase Mu 4, and glutathione S-transferase Mu 5, while a prior study highlighted glutathione S-transferase Mu 6. Additionally, we found cytochrome P450 2C3, 2C29, and 2C14, while a prior study highlighted cytochrome P450 2C41 [5]. The findings demonstrated the indispensability of utilizing the appropriate databases for protein identification. Furthermore, proteins associated with the basal metabolism were identified in our study, including galectin, ribosomal protein, glucose-6-phosphate isomerase, acyl-CoA-binding protein homolog, calmodulin, ATP synthase subunit beta, tubulin alpha chain, and hemoglobin subunit beta, as reported previously [6,21].

### 2.8. The Comparison of TV Proteins from Different Species

The cluster analysis of protein components in all TV samples was conducted in the form of a clustering heat map (Figure 3E). The results revealed four distinct clusters based on species, with samples from different regions within the same species tightly grouped together. The protein compositions of BaS and BgC were found to be quite similar, confirming their taxonomic relationships [15]. Although BrS did not exhibit the most distant taxonomic relationship with BgC, it showed the greatest disparity in protein composition compared to BgC.

Furthermore, differential protein markers were investigated for distinguishing *Bufo* species. Herein, 54 specific proteins were marked in the legal species. Additionally, 26 proteins were unique to BgC, 97 proteins were unique to BmS, 6 proteins were unique to BrS, and 73 proteins were unique to BaS (Figure 3F). Moreover, 49 proteins with high confidence were unique to legal species, 22 proteins with high confidence were unique to BgC, 86 proteins with high confidence were unique to BmS, 5 proteins with high confidence were unique to BrS, and 60 proteins with high confidence were unique to BaS, which as listed in Appendix A.

### 2.9. The Comparison of TV Proteins from Different Regions in the Same Species

There was minimal variation among the samples from different regions within the same species. The common proteins accounted for 82.36% of the detected proteins in BaS, 60.4% in BrS, and 76.38% in BgC (Figure 3G). The findings suggested that inter-sample variation within the same species was minimal, indicating that the species’ genetic makeup primarily determined the protein component profiles with a lesser influence from environmental factors and the drying process.

## 3. Discussion

This study aimed to identify the bufadienolide and protein profiles in 28 batches of TV samples from different *Bufo* species and to distinguish between legal species and counterfeits using HPLC, UPLC-Q-TOF/MS, SDS-PAGE, and proteomics analyses with Nano LC-MS/MS. The following discussions seek to interpret the interesting findings. Furthermore, this integrated quality evaluation approach can provide insights for future research on developing a robust quality control strategy for the distinction between the adulteration and counterfeiting of TV, as well as offering comprehensive means to evaluate both micromolecules and macromolecules in the alternative resources of TV.

### 3.1. Micromolecules in TV

Bufadienolides and indole alkaloids are two types of best-studied compounds generally considered as active constituents in TV [2,22]. Plenty of previous studies, dealing with a quality evaluation of TVs from different regions in China and the analysis of both bufadienolides and indole alkaloids by LC–MS/MS [9,18,23], provided a good basis for identifying chemical constituents in the present investigated samples. As a result, 126 compounds were identified, including 8 amino acids, 6 alkaloids, 103 bufadienolides (46 free and 57 conjugated bufadienolides), and 9 previously undetected components. The distribution of each identified compound in the tested TV samples, obtained by the semi-quantitative results from the extracted ion current chromatogram, demonstrated that BgC, BaS, and BrS displayed similarity in the number of components, whereas BmS was significantly different. Previous investigations reported that BgC and BmS had significantly different bufadienolides and indole alkaloids [9,10], and a recent publication systematically compared the metabolite profiles and antitumor activity of TVs derived from *B. gargarizans gargarizans* and *B. gararizans andrewsi* [24]. In contrast, this paper offers a comprehensive comparison between legal BgC and BmS, as well as the main confusion species, BaS, and BrS, based on the metabolomics and multi-component quantification. Especially for BrS, this is the first report on its bufadienolides and indole alkaloid profiles.

Most free and conjugated bufadienolides were identified in BgC, BaS, and BrS samples, where free bufadienolides displayed higher abundance. This is consistent with previous studies [9,16,18], but it was observed that the abundance of conjugated bufadienolides was higher in BmS samples. Due to the same processing methods utilized for different *Bufo* species, these abundant differences between the free and conjugated bufadienolides could be related to the living environment or the evolution of the toad. For free bufadienolides, various trends were observed regarding the absolute amount of main active compounds, CB and RB, in BaS samples. Our research showed that the total content of CB and RB in the BaS samples was similar to that in BgC, but the content of CB is higher than RB, which contradicted Sun’s findings [24]. Although all the BaS samples used in this research were from the Sichuan province in China, the different outcomes of CB and RB may still be caused by regional difference, and the quality of BaS can be further assessed by increasing the sample number in the future. Additionally, despite being a common confusion species in the trading market for years, BrS samples displayed significant discrepancy from BgC samples and exhibited a lesser content of bufadienolides. Therefore, it is imperative to conduct further activity evaluation and counterfeit identification regarding BrS.

### 3.2. Macromolecules in TV

The importance of proteins in TV has been increasingly emphasized [5]. Previous bioinformatic analysis has shown that the proteins in TV are involved in multiple pathways, such as the 5-HT-receptor-mediated signaling pathway, beta adrenergic receptor signaling pathway, and the biosynthesis of cofactors [4,5]. Although several studies have investigated proteins by electrophoretic analysis and LC-MS/MS, the emphasis is on TV from BgC collected from different geographical regions. However, there is still a gap in distinguishing different species by protein markers. Researchers from Brazil reported 42 identified proteins such as acyl-CoA-binding protein, alcohol dehydrogenase, calmodulin, galectin, and histone from the soluble protein fractions of the parotoid macro-gland secretion of *Duttaphrynus melanostictus*, namely BmS [6]. An irreversible serine protease inhibitor called baserpin, with an apparent molecular weight of about 60 kDa, and a BA-lysozyme with molecular weight of about 15 kDa, were purified from the skin secretions of *B. andrewsi* [25,26]. To date, there has still been no analysis of the protein composition profiles of BmS, BaS, and BrS samples. Herein, the protein components of four *Bufo* species were comprehensively analyzed employing the Bradford method, SDS-PAGE, and Nano LC-MS/MS in the present investigation.

The total protein content was significantly higher in the BgC and BaS samples compared to the BmS and BrS samples, with the majority exceeding 10%. The results were consistent with Sousa-Filho’s findings of 11–30% of secretion dry weight [21], which was substantially higher than the content of 6.63–19.42 mg/g described in the recent report [5]. Moreover, in accordance with previous reports that the most abundant protein band falls within the range of 38–60 kDa [5,27], our SDS-PAGE analysis unveiled a great distinct protein band among the four *Bufo* species, thereby indicating that SDS-PAGE analysis offers a straightforward and visually informative approach for discriminating between *Bufo* species. Additionally, 1357 proteins were identified in TV and some of them were bio-active proteins such as glucose-6-phosphate isomerase, catalase, and lysozyme C-1, which had catalytic, antioxidant, antimicrobial, and anti-tumor activities [4,5]. Furthermore, significant differences in protein expression were observed between the legal species BgC and BmS, as well as confusion species BaS and BrS based on protein identification. Our study presents the first comprehensive analyses of differences among the four *Bufo* species, notably investigating the characteristics specific to BrS for the first time.

### 3.3. Integrated Analysis Supported Taxonomic Relationship of Toads

The taxonomic status of BgC, BaS, and BmS has been controversial to date, which was one of the objective reasons for the existence of TV counterfeits. In the cases of BgC and BaS, these are usually classified as two independent species [28], while they are regarded as subspecies in *Fauna Sinica* (*amphibia*), and in Othman’s findings, they are even combined into one [29]. For BmS, these belonged to the genus *Bufo* in *Fauna Sinica* (amphibia), but other scholars reconsidered them as a species in the genus *Duttaphrynus* [30]. The present findings, combining micromolecules and proteins, supported the relatedness order that BgC is close to BaS, followed by BrS, and then far from BmS. If BgC and BaS were considered subspecies, BaS could be a legal alternative resource for TV due to their similarity. However, if it is classified as a separate species, BaS would be considered as a counterfeit since its identification method is yet to be established. Another counterfeit, BrS, belongs to the *B. raddei* group, which is closely related to the *B. gargarizans* groups, as shown in Figure 1.

Conversely, as one of the legal origin sources in ChP, BmS displayed distinct profiles compared to BgC based on our comprehensive analysis of micro- and macromolecules. The similar results were also visible from the taxonomic relationships depicted in Figure 1. Moreover, revealing the structure of undescribed components unique to BmS warrants further attention. These notable disparities between BmS and BgC, both in quality evaluation and taxonomic relationships, have raised our concern about the clinical effectiveness of BmS, and additional evidence is necessary to substantiate the suitability of BmS as a legal resource in future studies.

### 3.4. Future Research

TV, being a traditional medicine with significant therapeutic efficacy, is suffering from the increasing confusion about the source of medicinal materials due to the rising market demand and rising prices. Additionally, the migration of production from traditional regions to emerging ones has further contributed to the confusion of the *Bufo* species. The abuse of illegal *Bufo* resources may exacerbate the risks of clinical application. Despite the extensive studies on TV, there has been a lack of focus on differentiating between legally authorized species and mixed counterfeits. Therefore, the current status of TV urges stricter demands for quality control: it is imperative to explore more efficient and comprehensive means. According to the findings of our study, significant differences were observed, both in micromolecules and macromolecules, between BgC and BmS, the two legally recognized species. However, BaS, commonly used as a counterfeit substitute, exhibited a comparable quality level to that of BgC, and this perspective is supported by the taxonomic relationship. The approaches established in this study provide strategies for effectively distinguishing *Bufo* species, and the results offer scientific foundations for selecting high-quality species as alternative sources of TV. Furthermore, the next focus of our work would be to establish a more rational and practical method for TV quality assessment, as well as to explore the feasibility of alternative TV sources.

## 4. Conclusions

As one of the traditional toxic drugs, TV is widely used in the clinic and exhibits favorable clinical efficacy. In this article, an integrated quality evaluation strategy using HPLC, UPLC-Q-TOF-MS/MS, SDS-PAGE, and Nano LC-MS/MS was employed to comprehensively characterize and evaluate the chemical and protein constituents of TVs from different *Bufo* species. In total, 126 chemical compounds were identified in TV samples. The contents of the five representative bufadienolides and total protein content in TV samples from BgC and BaS were found to be higher than those from BmS and BrS. The SDS-PAGE and MS/MS analysis revealed a significant difference in proteins between BmS and the three other *Bufo* species, with distinct proteins identified for each species. Based on the taxonomic relationships, the content of chemical compounds and total protein contents, as well as qualitative analysis on micromolecules and proteins perspective, the TV samples from BgC and BaS were similar in terms of their superior quality. Compared to BmS, BaS and BrS are more reasonable alternative sources of TV. This research provided comprehensive evidence for developing quality evaluation approaches for TV, and offered helpful information for making rational decisions regarding better *Bufo* species as the origin animals of TV.

## 5. Materials and Methods

### 5.1. Materials

All reagents (analytical grade or higher) were purchased from Sigma-Aldrich Corporation (St. Louis, MO, USA), Thermo Fisher Scientific Corporation (Waltham, MA, USA), Roche Corporation (Basel, Switzerland), Promega Corporation (Madison, WI, USA), Sinopharm Chemical Reagent Co., Ltd. (Shanghai, China), Tianjin Guanfu Fine Chemicals Research Institute (Tianjin, China), and Beijing Zhongshan Jinqiao Bio-Tech Co., Ltd. (Beijing, China).

Chemical reference substances CB (110803-201406), RB (110718-201809), and BF (111981-201501) were obtained from the National Institutes for Food and Drug Control (Beijing, China). GB (HG4515S1) and BL (P28M10F84299) were obtained from Baoji Herbest Bio-Tech Co., Ltd (Baoji, China) and Shanghai Yuanye Bio-Tech Co., Ltd. (Shanghai, China), respectively. Bovine serum albumin was obtained from Sigma-Aldrich Corporation (St. Louis, MO, USA).

### 5.2. TV Samples

A total of 28 batches of four *Bufo* samples were collected from their representative geographical origins, including Jiangsu, Shandong, Shanxi, Gansu, Henan, Zhejiang, Guangxi, Jiangxi, Fujian, Heilongjiang, Jilin, and Sichuan. Among them, 16 batches of samples are BgC, 7 are BmS, 2 are BaS, and 3 are BrS. Five batches of samples, including BgC-F2, BgC-F16, BmS-F1, BaS-F1, and BaS-F2, were fresh venom, and another 23 batches of samples were dry venom. The fresh and dry samples were compared to reveal the effect of processing on TV. The calculated bufadienolide and protein content in five batches of fresh venom were obtained according to the actual measured values with a deduction of moisture content of 66%. All samples were identified by one of the co-authors, Professor Huimin Gao, and deposited at the Institute of Chinese Materia Medica, China Academy of Chinese Medical Sciences, Beijing, China. Sample details are provided in Table 2.

### 5.3. UPLC-Q-TOF/MS Analysis

Twenty-five milligrams of TV were extracted with twenty milliliters of methanol by ultrasonication for 30 min, and then centrifuged (10,625× *g*, 10 min) to obtain the supernatant. The analyses were carried out using an Xevo G2-S Q-TOF mass spectrometer (Waters, Milford, MA, USA) with an ESI source connected to an ACQUITY UPLC H-CLASS system (Waters, Milford, MA, USA).

An ACQUITY BEH C_18_ column (2.1 mm × 100 mm, 1.7 µm) (Waters, Milford, MA, USA) with a column temperature maintained at 30 °C was employed. The mobile phases were acetonitrile (ACN) (A) and water (B) with 0.1% formic acid. The gradient elution program was as follows: 0–8 min, 8–54% A; 8–10 min, 54% A; 10–13 min, 54–95% A. The flow rate was 0.4 mL/min, and the injection volume was 1 µL.

Mass spectra were acquired in positive ion modes. MS conditions were as follows: capillary, 1.8 kV; source temperature, 120 °C; drying gas flow, 5.0 L/min; cone gas flow, 50 L/h; collision energy, 6.0 eV; tube lens, 34 V. The mass spectrometer was operated within a range of *m*/*z* 100–2000. The compounds containing bufadienolides and alkaloids were identified by comparison of the fragmentation pattern, UV profile, and mass spectra.

### 5.4. Quantitative Analysis of Bufadienolides

A Shimadzu LC-20 AT system with SPD-M20A diode array detector (Kyoto, Japan) and a Grace Alltima C_18_ column (4.6 mm × 250 mm, 5 µm) was used for the HPLC analysis. The injection volume was 10 µL, and the column temperature was maintained at 30 °C. Twenty-five milligrams of TV was refluxed with twenty milliliters of methanol in a water bath for 1 h, and centrifuged (10,625× *g*, 10 min). The mobile phases were ACN (A) and water (B), containing 0.3% acetic acid in gradient elution mode. The elution program was as follows: 0–15 min, 28–54% A; 15–35 min, 54% A. The flow rate was 0.6 mL/min. The detection wavelength was set at 296 nm.

### 5.5. Determination of Total Proteins by Bradford Method

After 12.5 mg of TV was extracted with five milliliters of protein lysis buffer (constituted by 7 M Urea, 2 M Thiourea, 0.1% Triton X-100, 2% protease inhibitor cocktails, and 50 mM NH_4_HCO_3_) by ultrasonication (250 W, 40 kHz) for 30 min, it was centrifuged (13,390× *g*, 10 min, 4 °C) to obtain the supernatant. This was followed by proceeding the protocol of the Bradford reagent (Sigma-Aldrich, St. Louis, MO, USA), measuring an absorbance value at 596 nm by a Multiskan GO Microplate Spectrophotometer (Thermo, Waltham, MA, USA).

### 5.6. SDS-PAGE Analysis

Fifty milligrams of TV were extracted with two milliliters of protein lysis buffer by ultrasonication (250 W, 40 kHz) for 30 min, then centrifuged (13,390 × *g*, 10 min, 4 °C) to obtain the supernatant. The protein concentration was measured by Bradford method. Each sample (containing 30 µg protein) was mixed with 5× loading buffer and boiled for 5 min. Then, 5% (*w*/*v*) acrylamide stacking gels were used in combination with 12% (*w*/*v*) separating gels (1 mm thickness) prepared from 30% (*w*/*v*) acrylamide/bis acrylamide solution (*w*/*w*, 29:1) (Sigma-Aldrich, St. Louis, MO, USA) using a Mini-Protean gel casting and electrophoresis system (Bio-Rad, Hercules, CA, USA). Electrophoresis was conducted alongside a 10–250 kDa protein marker (Thermo Fisher Scientific, Waltham, MA, USA), and the gels were then washed with deionized water. The staining of the protein was performed using Coomassie brilliant blue dye for 60 min with gentle shaking. Then, the gels were washed 2 × 5 min with deionized water, and destained with 30% (*v*/*v*) methanol-10% (*v*/*v*) acetic acid until the background was sufficiently low.

### 5.7. In-Solution Digestion of Proteins

Fifty milligrams of TV were extracted with two milliliters of protein lysis buffer by ultrasonication (250 W, 40 kHz) for 30 min and centrifuged (13,390× *g*, 10 min, 4 °C) as described above. The lysates (containing 250 µg protein) were reduced with 10 mM dithiothreitol, followed by incubation for 3 h at 37 °C, then alkylated with 50 mM iodoacetamide at room temperature, in the dark, for 1 h. The urea concentration in the sample solution was reduced to 1 M by diluting the samples with 50 mM NH_4_HCO_3_. The proteins were digested with trypsin (Promega, Madison, WI, USA) overnight. The protein-to-enzyme ratio was 100:1. The next day, the enzyme was added again at the same ratio for 1 h. Then, protein digestion was stopped by adding formic acid at a final concentration of 0.1%. Digestion products were desalted by Sep-Pak^®^ C_18_ column (Waters, Milford, MA, USA), and then evaporated with a vacuum centrifugal concentrator (Eppendorf, Hamburg, Germany).

### 5.8. Nano LC-MS/MS Analysis and Protein Identification

A nanoflow UPLC instrument (EASY-nLC 1000 system, Thermo Fisher Scientific, Waltham, MA, USA) was coupled to a Q-Exactive mass spectrometer with a nano-electrospray ion source (Thermo Fisher Scientific, Waltham, MA, USA) for proteome analysis. The samples were separated on a Reprosil-Pur C_18_ AQ column (75 µm × 20 cm, 3 µm) (Dr. Maisch, Tübingen, BW, Germany). The mobile phase was water containing 0.1% formic acid (A) and ACN containing 0.1% formic acid (B). The gradient elution program was as follows: 0–8 min, 4–8% B; 8–58 min, 8–22% B; 58–70 min, 22–32% B; 70–71 min, 32–95% B; 71–78 min, 95% B. The flow rate was 300 nL/min. The electrospray voltage was 2.0 kV. The dynamic exclusion time was 40 s. In MS^1^, the resolution was 70,000, the scan range was 300–1600 *m*/*z*, the AGC target was 3 × 10^6^, and the maximum injection time was 60 ms. In MS^2^, the resolution was 17,500, the AGC target was 5 × 10^4^, and the maximum injection time was 80 ms. The 20 most intensive precursor ions were selected for MS/MS analysis.

The raw data were analyzed with Proteome Discoverer software (version 2.2, Thermo Fisher Scientific, Waltham, MA, USA) using Sequest HT search engine for identification and label free quantitation analysis. The cane toad proteins database [31] was used for searching the data from all samples. Searching parameters were set as follows: digestion Trypsin; a maximum number of missed cleavages of 2; fixed modification: cysteine carbamidomethylation; variable modification: methionine oxidation; mass tolerance of precursor ions: 10 ppm; mass tolerance of product ions: 0.02 Da; false discovery rate: 1%.

### 5.9. Statistical Analysis and Bioinformatic Analysis

The PCA analyses for *Bufo* species were conducted by Progenesis QI software (version 2.3, Waters, Milford, MA, USA). To explore the biological functions and related pathways of the identified proteins, DAVID Bioinformatics Resources was employed.

## Figures and Tables

**Figure 1 toxins-16-00159-f001:**
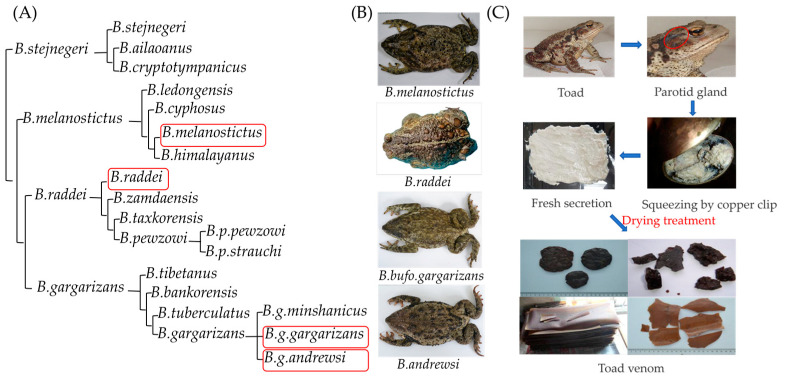
The taxonomic relationship and representative pictures of four *Bufo* species, and the workflow of harvesting and processing TV. (**A**) The taxonomic relationship between four *Bufo* species according to *Fauna Sinica*. The four red circles in (**A**) represent the latin name of four species in *Fauna Sinica*. (**B**) The representative pictures of four *Bufo* species (BmS, BrS, BgC, and BaS). (**C**) The workflow of harvesting and processing TV involves collecting secretion from parotid gland of Toad, drying it to obtain different forms of TV.

**Figure 2 toxins-16-00159-f002:**
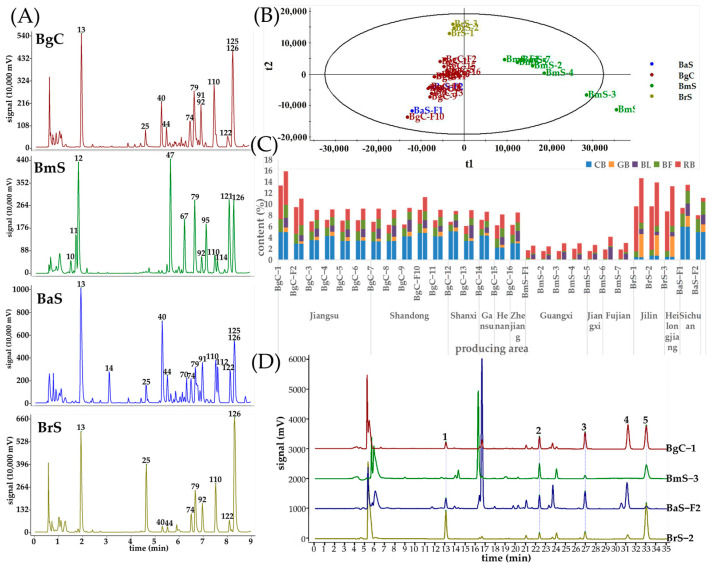
The analysis of the micromolecular differences between the four *Bufo* species. (**A**) The characteristics of four *Bufo* species obtained by UPLC-Q-TOF/MS processed by the fingerprint similarity software (Similarity Evaluation System for Chromatographic Fingerprint of Traditional Chinese Medicine, version 2012.130723). Peak numbering in accordance with numbering in Table 1. (**B**) The principal component analysis (PCA) score plot of 28 samples from four *Bufo* species based on UPLC-Q-TOF/MS. (**C**) The content of bufadienolides in 28 batches of TV samples: the left bar of each sample represents the total content of BF, CB, and RB; and the right bar of each sample represents the total content of GB, BF, BL, CB, and RB. (**D**) The typical HPLC chromatograms of TV from four species, and peaks 1–5 were gamabufotalin (GB), bufotalin (BL), bufalin (BF), cinobufagin (CB), and resibufogenin (RB), respectively. The numbering of the samples is available in Table 2.

**Figure 3 toxins-16-00159-f003:**
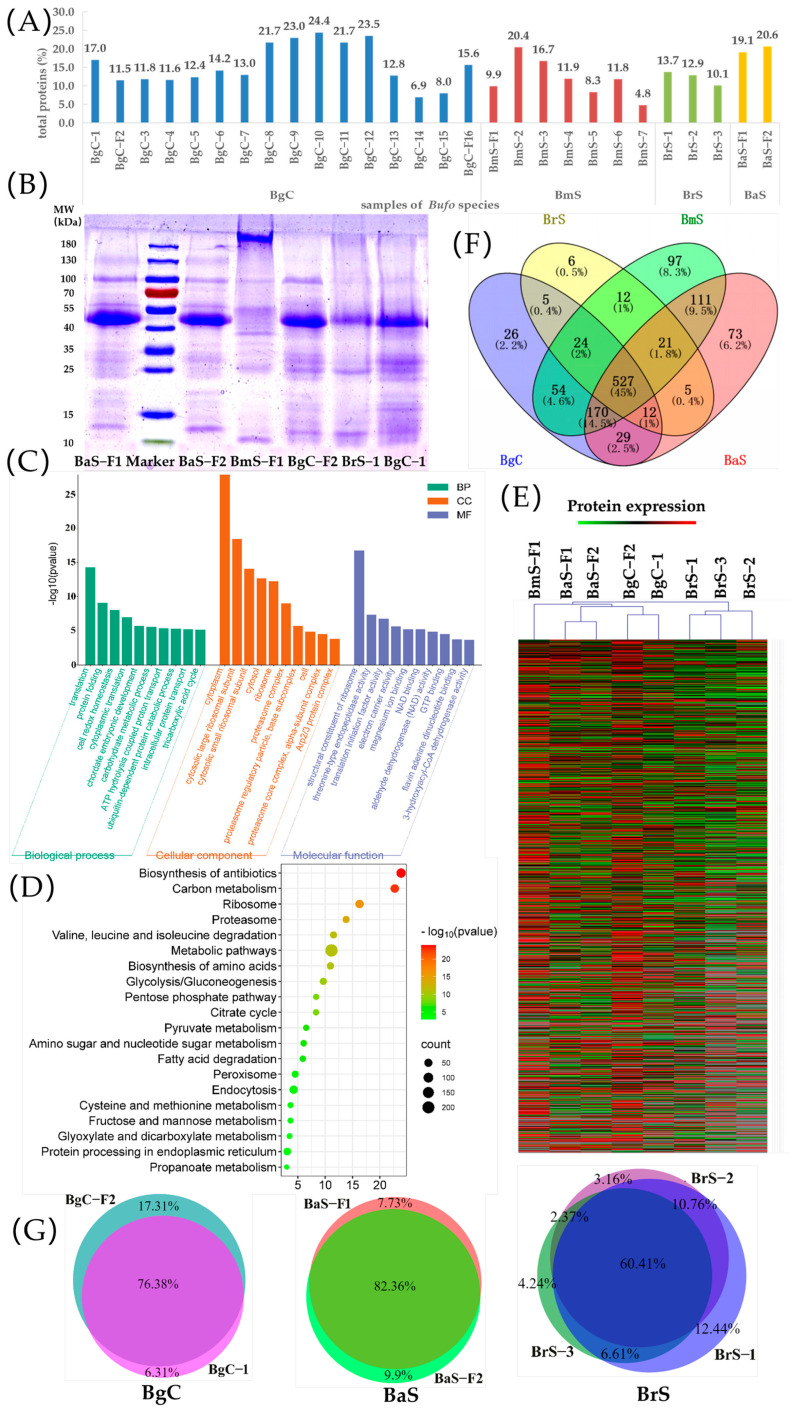
The protein differences among four *Bufo* species. (**A**) Total protein content of 28 samples. The blue, red, green, and yellow bars represented the BgC, BmS, BrS, and BaS samples, respectively. The number at the top of the column was the total protein content. (**B**) Sodium dodecylsulfate-polyacrylamide gel electrophoresis (SDS-PAGE) analysis of the typical samples of four *Bufo* species BaS, BmS, BgC, and BrS. The 5% (*w*/*v*) acrylamide stacking gels were used in combination with 12% (*w*/*v*) separating gels. (**C**) Gene Ontology (GO) enrichment analysis of all identified proteins in four *Bufo* species. The top ten terms of the three GO domains, namely biological process (BP), cellular component (CC), and molecular function (MF), are displayed. The y axis showed the value of −log10(*p* value) of each GO classification. (**D**) Kyoto Encyclopedia of Genes and Genomes (KEGG) pathway enrichment analysis of all identified proteins in four *Bufo* species. The x axis exhibited the value of −log10 (*p*-value) of each term. The y axis showed the top twenty terms. (**E**) Cluster analysis of the similarity of proteins in four *Bufo* species. (**F**) Venn diagram of the proportion of common and different proteins in the four *Bufo* species. The purple, yellow, green, and red parts represented BgC, BrS, BmS, and BaS samples, respectively. (**G**) The differences of proteins among samples of the same species from different regions. The different colors represent the BgC-1, BgC-F2, BaS-F1, BaS-F1, BrS-1, BrS-2 and BrS-3 samples.

**Table 1 toxins-16-00159-t001:** The 126 compounds in the TVs from different *Bufo* species.

No.	Name	RT (min)	[M+H]^+^ Detected	[M+H]^+^ Expected	Error (ppm)	Formula	*Bufo* Species ^1^			
							BgC	BmS	BrS	BaS
1	succinyl arginine	0.60	275.1359	275.1355	1.5	C_10_H_18_N_4_O_5_	++++	−	++++	++++
2	adipyl arginine	0.79	303.1674	303.1668	2.0	C_12_H_22_N_4_O_5_	+++	−	+++	++++
3	serotonin	0.89	177.1028	177.1028	0.0	C_10_H_12_N_2_O	+	−	−	+
4	N-methyl serotonin	0.97	191.1185	191.1184	0.5	C_11_H_14_N_2_O	++	++	++	++
5	N,N-dimethyl serotonin	1.03	205.1345	205.1341	1.9	C_12_H_16_N_2_O	++	+	+++	++
6	N,N,N-trimethyl serotonin	1.04	219.1497	219.1497	0.0	C_13_H_18_N_2_O	+++	+++	+++	++++
7	pimeloyl arginine	1.12	317.1830	317.1825	1.6	C_13_H_24_N_4_O_5_	+++	+	+++	++++
8	dehydrobufotenine	1.30	203.1188	203.1184	2.0	C_12_H_14_N_2_O	+++	++	+++	+++
9	bufothionine	1.40	283.0751	283.0753	−0.7	C_12_H_14_N_2_O_4_S	++	++	++	++
10	suberoyl-L-histidine	1.53	312.1562	312.1559	1.0	C_14_H_21_N_3_O_5_	+	+++	+	+
11	suberoyl-L-1-methylhistidine	1.73	326.1717	326.1716	0.3	C_15_H_23_N_3_O_5_	++	++++	++	+++
12	unknown	1.82	245.1866	245.1865	0.4	C_12_H_24_N_2_O_3_	+	++++	++	−
13	suberoyl arginine	1.95	331.1984	331.1981	0.9	C_14_H_26_N_4_O_5_	++++	++	++++	++++
14	sebacyl arginine isomer	2.42	359.2296	359.2294	0.6	C_16_H_30_N_4_O_5_	+++	++	+++	+++
15	unknown	2.85	416.2432 ^2^	416.2437	−1.2	C_24_H_30_O_5_	+	−	+	+
16	azelayl arginine	3.12	345.2142	345.2138	1.2	C_15_H_28_N_4_O_5_	+	−	+	++++
17	11α-hydroxyltelocinobufagin	3.95	419.2418	419.2434	−3.8	C_24_H_34_O_6_	++	++	+	++
18	19-hydroxyltelocinobufagin	4.22	419.2423	419.2434	−2.6	C_24_H_34_O_6_	++	+	++	++
19	5,12β-dihydroxycinobufagin	4.23	475.2322	475.2332	−2.1	C_26_H_34_O_8_	+	++	+	+
20	ψ-bufarenogin	4.44	417.2277	417.2277	0.0	C_24_H_32_O_6_	+++	++	++	+++
21	gamabufotalin 3-O-succinoyl arginine ester or its isomer	4.48	656.3658	659.3656	0.3	C_34_H_50_N_4_O_9_	+	−	++	+
22	16-acetoxybufarenogin	4.53	475.2328	475.2332	−0.8	C_26_H_34_O_8_	+	++	+	+
23	16β-Hydroxyl-pseudobufarenogin	4.56	433.2215	433.2226	−2.6	C_24_H_32_O_7_	−	−	+	++
24	3-oxo-12β-hydroxyl desacetylcinobufagin	4.56	415.2111	415.2121	−2.4	C_24_H_30_O_6_	−	−	+	++
25	gamabufotalin	4.65	403.2485	403.2484	0.2	C_24_H_34_O_5_	+++	++	++++	++++
26	bufarenogin	4.72	417.2272	417.2277	0.0	C_24_H_32_O_6_	+++	++	++	+++
27	11α,19-dihydroxylmarinobufagin	4.75	433.2221	433.2226	−1.2	C_24_H_32_O_7_	−	−	+	++
28	gamabufotalin 3-O-succinoyl arginine ester or its isomer	4.76	659.3663	659.3656	1.1	C_34_H_50_N_4_O_9_	+	−	++	+
29	16-O-acetylarenobufagin	4.77	475.2334	475.2332	0.4	C_26_H_34_O_8_	++	+++	++	++
30	1β-hydroxylbufalin	4.78	403.2476	403.2484	−2.0	C_24_H_34_O_5_	++	++	−	+++
31	gamabufotalin 3-O-succinoyl arginine ester or its isomer	5.08	659.3654	659.3656	−0.3	C_34_H_50_N_4_O_9_	+	−	++	+
32	arenobufagin/hellebrigenin 3-O-succinoyl arginine	5.15	673.3824	673.3813	1.6	C_35_H_52_N_4_O_9_	−	−	−	+
33	arenobufagin 3-O-adipoyl arginine ester	5.19	701.3748	701.3762	−2.0	C_36_H_52_N_4_O_10_	+	−	−	+++
34	19-oxo-desacetylcinobufagin	5.19	415.2129	415.2121	1.9	C_24_H_30_O_6_	++	−	−	++
35	hellebrigenol	5.19	419.2427	419.2434	−1.7	C_24_H_34_O_6_	++	++	+	++
36	hellebrigenol 3-O-suberoyl arginine ester or its isomer	5.21	731.4226	731.4231	−0.7	C_38_H_58_N_4_O_10_	+	−	+	+++
37	1β-Hydroxylarenobufagin	5.23	433.2211	433.2226	−3.5	C_24_H_32_O_7_	−	−	+	+
38	gamabufotalin isomer	5.28	403.2479	403.2484	−1.2	C_24_H_34_O_5_	++	++	+	+++
39	5-hydroxy bufotalin	5.28	461.2531	461.2539	−1.7	C_26_H_36_O_7_	++	++	+	++
40	arenobufagin	5.31	417.2281	417.2277	1.0	C_24_H_32_O_6_	++++	+++	+++	++++
41	cinobufaginol isomer	5.42	459.2371	459.2383	−2.6	C_26_H_34_O_7_	++	++	+	++
42	hellebrigenin 3-O-adipoyl arginine ester	5.48	701.3774	701.3762	1.7	C_37_H_56_N_4_O_9_	+	−	+	+++
43	arenobufagin/hellebrigenin 3-O-pimeloyl arginine ester	5.51	715.3908	715.3918	−1.4	C_37_H_54_N_4_O_10_	−	−	−	++
44	hellebrigenin	5.52	417.2276	417.2277	−0.2	C_24_H_32_O_6_	++++	+++	+++	++++
45	gamabufotalin 3-O-adipoyl arginine ester or its isomer	5.58	687.3976	687.3969	1.0	C_36_H_54_N_4_O_9_	−	+	−	−
46	hellebrigenol 3-O-suberoyl arginine ester or its isomer	5.66	731.4214	731.4231	−2.3	C_38_H_58_N_4_O_10_	+	−	+	++
47	19-hydroxybufalin	5.68	403.2494	403.2484	2.5	C_24_H_34_O_5_	+++	++++	++	+++
48	gamabufotalin 3-O-adipoyl arginine ester or its isomer	5.76	687.3953	687.3969	−1.9	C_36_H_54_N_4_O_9_	+	−	+	++
49	5β-Hydroxyl-14α-artebufogenin	5.78	401.2325	401.2328	−0.7	C_24_H_32_O_5_	++	++	+	+++
50	cinobufaginol	5.80	459.2391	459.2383	1.7	C_26_H_34_O_7_	+++	−	++	+++
51	monohydroxylbufotalin	5.81	461.2528	461.2539	−2.4	C_26_H_36_O_7_	++	+	++	+++
52	hellebrigenin/arenobufagin 3-O-suberoyl arginine ester or its isomer	5.88	729.4062	729.4075	−1.8	C_38_H_56_N_4_O_10_	++	−	+	++++
53	unknown	5.91	615.4006	615.4009	−0.5	C_35_H_54_N_2_O_7_	−	++	−	−
54	gamabufotalin 3-O-suberoyl arginine ester or its isomer	5.92	715.4272	715.4282	−1.4	C_38_H_58_N_4_O_9_	++	−	+++	+++
55	bufotalinin	5.92	415.2113	415.2121	−1.9	C_24_H_30_O_6_	++	++	+++	++
56	desacetylbufotalin	6.00	403.2481	403.2484	−0.7	C_24_H_34_O_5_	+++	+++	++	+++
57	resibufaginol	6.03	401.2324	401.2328	−1.0	C_24_H_32_O_5_	++	+++	+++	++
58	desacetylcinobufaginol	6.03	417.2255	417.2277	−5.3	C_24_H_32_O_6_	−	−	−	+
59	19-oxo-cinobufotalin	6.05	473.2178	473.2175	0.6	C_26_H_32_O_8_	+++	−	++	+++
60	bufotalinin 3-O-suberoyl arginine ester	6.11	727.3909	727.3918	−1.2	C_38_H_54_N_4_O_10_	+	−	+	++
61	gamabufotalin 3-O-pimeloyl arginine ester or its isomer	6.12	701.4124	701.4126	−0.3	C_37_H_56_N_4_O_9_	++	−	+	+++
62	1β-hydroxylcinobufagin	6.12	459.2368	459.2383	−3.3	C_26_H_34_O_7_	−	−	−	−
63	argentinogenin	6.15	415.2115	415.2121	−1.4	C_24_H_30_O_6_	++	+	+	++
64	19-oxo-cinobufotalin 3-O-suberoyl arginine ester	6.15	785.3965	785.3973	−1.0	C_40_H_56_N_4_O_12_	+	−	−	+++
65	bufalin 3-O-succinoyl arginine ester or its isomer	6.16	643.3726	643.3707	3.0	C_34_H_50_N_4_O_8_	+++	+	++	++++
66	19-oxo-bufalin	6.23	401.2323	401.2328	−1.2	C_24_H_32_O_5_	+++	+++	++	+++
67	unknown	6.26	629.4162	629.4139	3.7	C_32_H_52_N_8_O_5_	−	++++	−	−
68	19-oxo-cinobufotalin-3-suberate methylhistidine	6.26	710.4012	710.4017	−0.7	C_34_H_50_N_3_O_9_	−	+++	−	−
69	gamabufotalin 3-O-suberoyl arginine ester or its isomer	6.30	715.4261	715.4282	−2.9	C_38_H_58_N_4_O_9_	+	−	+	++
70	hellebrigenin/arenobufagin 3-O-suberoyl arginine ester or its isomer	6.34	729.4077	729.4075	0.3	C_38_H_56_N_4_O_10_	+++	−	+	++++
71	cinobufaginol 3-O-suberoyl arginine ester or its isomer	6.38	771.4172	771.4180	−1.0	C_40_H_58_N_4_O_11_	++	−	+	+++
72	bufalin 3-O-glutaryl arginine ester or its isomer	6.45	657.3846	657.3863	−2.6	C_35_H_52_N_4_O_8_	++	−	+	+++
73	telocinobufagin 3-O-suberoyl arginine ester	6.50	715.4296	715.4282	2.0	C_38_H_58_N_4_O_9_	++	−	++	++++
74	telocinobufagin	6.52	403.2480	403.2484	−1.0	C_24_H_34_O_5_	++++	++	+++	++++
75	12β-hydroxylcinobufagin	6.57	459.2376	459.2383	−1.5	C_26_H_34_O_7_	++	−	+	+++
76	bufotalin 3-O-suberoyl-L-histidine or its isomer	6.62	738.3953	738.3966	−1.8	C_40_H_55_N_3_O_10_	−	+	−	−
77	resibufogenin 3-O-succinyl arginine ester or its isomer	6.66	641.3535	641.3550	−2.3	C_34_H_48_N_4_O_8_	++	−	++	+
78	desacetylcinobufagin	6.69	401.2325	401.2328	−0.7	C_24_H_32_O_5_	+++	+	++	+++
79	bufotalin	6.69	445.2589	445.2590	−0.2	C_26_H_36_O_6_	++++	++++	++++	++++
80	unknown	6.74	627.3989	627.4009	−3.2	C_36_H_54_N_2_O_7_	−	+++	−	+
81	cinobufaginol 3-O-suberoyl arginine ester or its isomer	6.74	771.4191	771.4180	1.4	C_40_H_58_N_4_O_11_	++	−	+	++++
82	desacetylcinobufagin 3-O-suberoyl arginine ester or its isomer	6.74	713.4122	713.4126	−0.6	C_38_H_56_N_4_O_9_	+	−	+++	+++
83	cinobufagin 3-O-succinoyl arginine ester or its isomer	6.76	699.3596	699.3605	−1.3	C_36_H_50_N_4_O_10_	+++	−	+	+++
84	bufalin 3-O-adipoyl arginine ester	6.81	671.4021	671.4020	0.1	C_36_H_54_N_4_O_8_	++	−	+	++++
85	resibufagin	6.82	399.2173	399.2171	0.5	C_24_H_30_O_5_	+++	+	+++	+++
86	19-oxo-cinobufagin	6.89	457.2230	457.2226	0.9	C_26_H_32_O_7_	+++	+	++	++++
87	resibufogenin 3-O-glutaryl arginine ester or its isomer	6.92	655.3706	655.3707	−0.2	C_35_H_50_N_4_O_8_	−	−	++	−
88	telocinobufagin 3-O-suberoyl arginine ester isomer	6.95	715.4286	715.4282	1.4	C_38_H_58_N_4_O_9_	++	−	+	+++
89	3-oxo-Δ^4^-resibufogenin	6.98	381.2062	381.2066	−1.0	C_24_H_28_O_4_	++	−	+	++
90	3-keto-cinobufagin	6.98	441.2273	441.2277	−0.9	C_26_H_32_O_6_	++	−	+	++
91	cinobufotalin	7.00	459.2381	459.2383	−0.4	C_26_H_34_O_7_	++++	−	+++	++++
92	marinobufagin	7.00	401.2323	401.2328	−1.2	C_24_H_32_O_5_	+++	+++	++++	++
93	bufotalin 3-O-suberoyl-L-histidine or its isomer	7.00	738.3961	738.3966	−0.7	C_40_H_55_N_3_O_10_	−	++	−	−
94	bufalin 3-O-suberoyl arginine ester or its isomer	7.01	699.4313	699.4333	−2.9	C_38_H_58_N_4_O_8_	−	−	−	+
95	unknown	7.19	671.4271	627.4245	3.9	C_34_H_54_N_8_O_6_	−	++++	−	−
96	resibufagin 3-O-suberoyl arginine ester or its isomer	7.07	711.3968	711.3969	−0.1	C_38_H_54_N_4_O_9_	+	−	+	++
97	bufotalin 3-O-suberoyl-L-3-methyl histidine	7.12	752.4113	752.4122	−1.3	C_41_H_57_N_3_O_10_	−	++	−	+
98	19-hydroxybufalin 3-O-suberoyl-L-histidine or its isomer	7.14	680.3892	680.3911	−2.8	C_38_H_53_N_3_O_8_	+	+	−	−
99	19-oxo-cinobufagin 3-O-suberoyl arginine ester or its isomer	7.14	769.4028	769.4024	0.5	C_40_H_56_N_4_O_11_	++	−	+	+++
100	bufotalin 3-O-suberoyl arginine ester or its isomer	7.17	757.4387	757.4388	−0.1	C_40_H_60_N_4_O_10_	++	−	++	+++
101	19-hydroxybufalin 3-O-suberoyl-L-3-methyl histidine or its isomer	7.17	694.4036	694.4067	−4.5	C_39_H_55_N_3_O_8_	−	+	−	−
102	bufalin 3-O-pimeloyl arginine ester or its isomer	7.19	685.4171	685.4176	−0.7	C_37_H_56_N_4_O_8_	+++	−	++	+++
103	marinobufagin 3-O-suberoyl arginine ester or its isomer	7.20	713.4130	713.4126	0.6	C_38_H_56_N_4_O_9_	+	−	+	++
104	resibufogenin 3-O-adipoyl arginine ester or its isomer	7.27	669.3846	669.3863	−2.5	C_36_H_52_N_4_O_8_	+	+	+	++
105	cinobufagin 3-O-adipoyl arginine ester or its isomer	7.30	727.3907	727.3918	−1.5	C_38_H_54_N_4_O_10_	++	−	+	+++
106	marinobufagin 3-O-suberoyl arginine ester or its isomer	7.38	713.4116	713.4126	−1.4	C_38_H_56_N_4_O_9_	−	−	+	+
107	cinobufotalin 3-O-suberoyl arginine ester	7.40	771.4157	771.4180	−3.0	C_40_H_58_N_4_O_11_	+	−	−	++
108	19-hydroxybufalin 3-O-suberoyl-L-histidine or its isomer	7.45	680.3909	680.3911	−0.3	C_38_H_53_N_3_O_8_	−	+	−	−
109	resibufogenin 3-O-suberoyl arginine ester or its isomer	7.48	697.4169	697.4176	−1.0	C_38_H_56_N_4_O_8_	+	−	+	++
110	bufalin	7.55	387.2531	387.2535	−1.0	C_24_H_34_O_4_	++++	+++	++++	++++
111	19-hydroxybufalin 3-O-suberoyl-L-3-methyl histidine or its isomer	7.61	694.4064	694.4067	−0.4	C_39_H_55_N_3_O_8_	−	++	−	++
112	bufalin 3-O-suberoyl arginine ester or its isomer	7.62	699.4321	699.4333	−1.7	C_38_H_58_N_4_O_8_	+++	+	+++	++++
113	resibufogenin 3-O-pimeloyl arginine ester or its isomer	7.67	683.4014	683.4020	−0.9	C_37_H_54_N_4_O_8_	+	−	++	−
114	unknown	7.64	613.4211	613.4217	−1.0	C_36_H_56_N_2_O_6_	−	+++	+	−
115	unknown	7.69	597.3895	597.3804	−1.5	C_35_H_52_N_2_O_6_	−	+++	−	−
116	cinobufagin 3-O-pimeloyl arginine ester or its isomer	7.74	741.4074	741.4075	−0.1	C_39_H_56_N_4_O_10_	++	−	+	+++
117	hellebrigenin 3-O-hemisuberate	7.85	573.3052	573.3064	−2.1	C_32_H_44_O_9_	+	−	+	++
118	resibufogenin 3-O-suberoyl-L-1-methyl histidine or its isomer	8.09	692.3909	692.3911	−0.3	C_39_H_53_N_3_O_8_				
119	bufalin 3-O-sebacyl arginine ester or its isomer	7.87	727.4630	727.4646	−2.2	C_40_H_62_N_4_O_8_	++	−	+	++
120	resibufogenin 3-O-suberoyl arginine ester or its isomer	8.10	697.4158	697.4176	−2.8	C_38_H_56_N_4_O_8_	++	−	+++	+++
121	unknown	8.12	611.4075	611.4073	0.3	C_37_H_50_N_6_O_2_	−	++++	++	−
122	cinobufagin 3-O-suberoyl arginine ester	8.14	755.4223	755.4231	−1.1	C_40_H_58_N_4_O_10_	+++	−	++	++++
123	gamabufotalin 3-O-hemisuberate	8.16	559.3248	559.3271	−4.1	C_32_H_46_O_8_	+	−	++	++
124	22,23-Epoxyresibufogenin	8.32	401.2332	401.2328	1.0	C_24_H_32_O_5_	++	−	+	+
125	cinobufagin	8.33	443.2433	443.2434	−0.2	C_26_H_34_O_6_	++++	+	+++	++++
126	resibufogenin	8.33	385.2377	385.2379	−0.5	C_24_H_32_O_4_	++++	++++	++++	+++

^1^ If the component was identified in more than 2/3 of all samples of each *Bufo* species, it was considered present in this species. The number of “+” indicates the response value of the ion chromatogram: “+” indicates that the level of e^3^ is reached; “++” indicates that the level of e^4^ is reached; “+++” indicates that the level of e^5^ is reached; and “++++” indicates that the level of e^6^ is reached. ^2^ *m*/*z* 416.2432 is [M+NH_4_]^+^.

**Table 2 toxins-16-00159-t002:** The contents of five bufadienolides and total proteins in 28 batches of TV samples.

No.	Source	Content ^1^ (%)							
		CB	GB	BL	BF	RB	BF+CB+RB	CB+GB+BL+BF+RB	Total Proteins
BgC-1	Nantong, Jiangsu	4.97	0.89	1.69	2.39	5.99	13.35	15.93	17.0
BgC-F2	Haimen, Jiangsu	2.94	0.38	1.15	1.75	4.80	9.49	11.02	11.5
BgC-3	Xuzhou, Jiangsu	3.52	0.62	1.45	1.61	1.80	6.93	9.00	11.8
BgC-4	Rugao, Jiangsu	4.28	0.82	1.05	1.57	1.36	7.21	9.08	11.6
BgC-5	Shanxian, Shandong	4.27	0.81	1.04	1.57	1.31	7.15	9.00	12.4
BgC-6	Baoying, Jiangsu	3.43	0.69	1.35	1.65	2.01	7.09	9.13	14.2
BgC-7	Linyi, Shandong	3.35	0.50	1.79	1.72	1.82	6.89	9.18	13.0
BgC-8	Taicang, Jiangsu	3.48	0.60	1.49	1.64	2.01	7.13	9.22	21.7
BgC-9	Xianyang, Shanxi	5.13	0.73	1.18	1.17	0.59	6.89	8.80	23.0
BgC-10	Longnan, Gansu	4.38	0.47	1.01	1.16	1.65	7.19	8.67	24.4
BgC-11	Linyi, Shandong	3.44	0.73	1.99	1.36	1.51	6.31	9.03	21.7
BgC-12	Dezhou, Shandong	4.25	0.80	2.15	1.42	0.74	6.41	9.36	23.5
BgC-13	Linfen, Shanxi	3.40	0.46	2.39	1.22	1.45	6.07	8.92	12.8
BgC-14	Zhoukou, Henan	2.23	0.51	1.38	1.58	2.45	6.26	8.15	6.9
BgC-15	Ningbo, Zhejiang	2.96	0.36	1.86	1.54	1.83	6.33	8.55	8.0
BgC-F16	Linyi, Shandong	4.85	0.86	1.41	1.64	2.50	8.99	11.26	15.6
BmS-F1	Guilin, Guangxi	0.12	0.04	0.80	0.25	1.36	1.73	2.57	9.9
BmS-2	Pingnan, Guangxi	-	0.04	0.79	0.22	1.40	1.62	2.45	20.4
BmS-3	Guilin, Guangxi	-	0.02	1.37	0.20	1.43	1.63	3.02	16.7
BmS-4	Guilin, Guangxi	-	0.05	0.91	0.21	1.81	2.02	2.98	11.9
BmS-5	Ganzhou, Jiangxi	-	0.02	1.09	0.20	1.41	1.61	2.72	8.3
BmS-6	Zhangzhou, Fujian	-	0.02	2.26	0.22	1.64	1.86	4.14	11.8
BmS-7	Wuyishan, Fujian	-	0.03	1.12	0.23	1.69	1.92	3.07	4.8
BrS-1	Yanji, Jilin	0.47	4.19	0.89	1.10	8.03	9.60	14.68	13.7
BrS-2	Wuchang, Heilongjiang	0.55	3.70	0.78	1.06	7.12	8.73	13.21	12.9
BrS-3	Huadian, Jilin	0.80	3.61	0.68	1.00	7.81	9.61	13.90	10.1
BaS-F1	Chengdu, Sichuan	5.97	1.92	2.20	2.33	1.08	9.38	13.50	19.1
BaS-F2	Yibin, Sichuan	4.98	1.40	1.71	2.38	0.68	8.04	11.15	20.6

^1^ CB: cinobufagin; GB: gamabufotalin; BL: bufotalin; BF: bufalin; RB: resibufogenin.

## Data Availability

The mass spectrometry proteomics data were deposited to the Proteome X change Consortium via the iProX partner repository with the dataset identifier PXD048668.

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
