# Peer review of "Comprehensive Analysis of Bufadienolide and Protein Profiles of Gland Secretions from Medicinal Bufo Species"

_toxins, 2024, doi:10.3390/toxins16030159_

Round 1

Reviewer 1 Report

Comments and Suggestions for Authors

 The co-authors describe the analysis of bufadienolide and protein profiles obtained from the gland secretion of Bufo species. In this research, a study is carried out with various techniques, trying to cover an analysis that is as complete as possible given the importance of the species of the Bufo genus. In order for the manuscript to be published, it is essential to address the following comments:

In the summary, include the most outstanding quantitative data from the research.

Improve the resolution of the figures

Note references from Table S2

The supplementary tables indicate the scores supporting high confidence, in addition to the program used to calculate the pI.

Table S1: attach the meaning of the abbreviations at the bottom.

Figures 2 and 4 indicate based on which table or figure the comparison is made.

Figure 3B: If an MS analysis is available, including an SDS PAGE analysis is unnecessary. However, the gel does not have a resolution that allows proteins to be identified or compared by band densities, in addition to proteins with molecular weights greater than 200 and less than ten kDa not being present. Suppose an analysis is performed with electrophoresis in denaturing gels. In that case, a gel with excellent resolution must be presented, and at least one where polypeptides of up to 250 kDa and 3 kDa can be visualized.

In the method, indicate the sonication conditions. Due to the sonication time, although the intensity used in this step is not indicated, how can it be explained that some proteins were not degraded under these sonication conditions?

In the manuscript, include the permission number for collecting the organisms and send the authorization from your institution's bioethics committee.

Comments on the Quality of English Language

It is necessary to check some grammar errors.

Reviewer 2 Report

Comments and Suggestions for Authors

Review comments for the research manuscript titled "Comprehensive Analysis on Bufadienolide and Protein Profiles of the Gland Secretions from Medicinal Bufo species" from the toxins journal.

The manuscript presents an intriguing study on the enhancement of toad venom quality and the selection of appropriate Bufo species. However, there are a few comments to be perperly addressed as follows.

1.      Improvements should be made to the quality of Figure images, especially for the resolutions of Fig 1 and Fig 2.

 2.      In "Materials and Methods" section, it is needed to clarify the methods of venom isolation in detail. For instance, what were the strength or intensity of the sonication applied to samples during ultrasonication, and what were the centrifugation speeds in g or rpm provided for the procedures in this section. In addition, the concentration of the protein loaded onto the column was not specified in the information provided. Only the total amount of venom used for the extraction was mentioned. Further, the description of the experimental procedures appear to be mixed up and not step-wise according to the experiment.

 3.      In line 495, the use of 25 mg of toad venom in the UPLC-Q-TOF/MS analysis raises questions. Given the sensitivity of the instrument, why was such a high load of venom used?

 4.      The venom isolation protocols differ in "Materials and Methods," particularly in "5.3 UPLC-Q-TOF/MS analysis," "5.5 Determination of total proteins by Bradford method," and "5.6 SDS-PAGE analysis." What accounts for these discrepancies?

 5.      When acronym was used, it should be written in full name at the very first time appeared in the manuscript. However, the manuscript does not follow the rule, for example, “UPLC-Q-TOF/MS”.

 6.      Figure 2 lacks a Y-axis in the chromatograms of UPLC-Q-TOF/MS and HPLC. This omission makes it difficult to properly visualize the intensity of the peaks.

 7.      In Figure 3, out of the total 28 TV samples from four Bufo species, they have used 6 samples for SDS-PAGE (Fig.3 B) and 8 samples for Cluster analysis (Fig.3 E). It's crucial to maintain consistency in reporting. If you intended to show 8 samples in protein expression but only presented 6 in the SDS page, it's important to explain any discrepancies. This could be due to technical issues, quality of samples, or other considerations.

1) The selection of the TV samples analyzed in these figures appears to be at random and those need to be addressed how they have chosen them from odd numbers of TV samples from four Bufo species.

2) On the other hand, they have analyzed only 3 species among the total four species included in the present study. Why is BmS species excluded in the analysis?

 8.      In the footnote of Table 1 (Lines 112- 114), the presence of "++++" symbols in the table data has not been properly addressed in the footnote. There are only “+, ++, +++” in the footnote. It can be an oversight or a typo and should be edited correctly.

Overall, this footnote paragraph is very obscured and hard to understand the meaning.

9.  In Materials and Methods, the concentration of the protein loaded onto the column was not specified in the information provided. Only the total amount of venom used for the extraction was mentioned. 

Round 2

Reviewer 1 Report

Comments and Suggestions for Authors

The manuscript is well understood, but the grammar must be corrected to clarify the entire message. In order to be published, it is necessary to attend to the following:

Line 5. "products" change for "product"

Line 7. "number" change for "a number"

Line 8. "To develop" change to "In order to develop"

Line 13-14. "Totally 126 compounds consisting of free or conjugated bufadienolides, indole alkaloids, and amino 15 acids were identified across the four Bufo species" change for "We identify 126 compounds of free or conjugated bufadienolides, indole alkaloids, and amino 15 acids in the "four Bufo species"

Line 15-16. "Differential proteins were also observed to distinguish Bufo species" change for "There are differences between the protein composition in the sample from the four Bufo species."

Line 19. "poor quality overall" change to "poorly overall"

Line 20. "This research provides evidence for developing approaches to evaluate TV quality and selecting the 21 proper Bufo species as origin source of TV listed in the Chinese pharmacopoeia." Change to "This research provides evidence for developing approaches to evaluate TV quality and selecting the 21 proper Bufo species as the origin source of TV listed in the Chinese pharmacopoeia."

In this sense, only the grammar of the summary was done, but this part needs to be improved. In this section of the summary, quantitative data that contribute substantially to the conclusions are requested.

Line 220-221. The Bradford assay is correct if "content" refers to the "total amount" of protein. However, in Table 2, the total amount of protein is not indicated; only a column indicates "total proteins." There is no data from the Bradford assay in Table 2.

Figure 3A refers to the Bradford assay. Again, this assay does not show total protein; the information it provides is the total amount of protein expressed in units of mass or mass/volume. The presentation of results for the Bradford trial is unclear.

In Figure 3B, gel analysis is not shown, only SDS-PAGE electrophoresis. In the text of the figure, include the percentage of acrylamide in the gel.

Line 249-250. They comment that there is "a little variation" in the leading bands of BgC and BaS TV; there are two lanes of BgC and two of BaS. What is the difference between the BgC samples? And what is the difference between the BaS samples? The gel is not of good quality, so a detailed analysis, particularly of the less abundant bands, cannot be performed. The SDS-PAGE electrophoresis experiment needs to be improved.

Line 501. Convert rpm to xg, or provide data on the rotor used. 

Comments on the Quality of English Language

Throughout the manuscript, there are numerous grammatical errors. It is imperative to send the manuscript to a Language Editing Service or a person whose native language is English. Some phrases are not very well understood due to grammatical errors.
